# The Dietary Effects of Nutmeg (*Myristica fragrans*) Extract on Growth, Hematological Parameters, Immunity, Antioxidant Status, and Disease Resistance of Common Carp (*Cyprinus carpio*) against *Aeromonas hydrophila*

Ghasem Rashidian [1],[†] , Khalid Shahin [2],[†] , Gehad E. Elshopakey [3] , Heba H. Mahboub [4] , Azin Fahim [1] , Hiam Elabd [5], Marko D. Prokić [6] and Caterina Faggio [7],*

1 Department of Aquaculture, Faculty of Natural Resources and Marine Sciences, Tarbiat Modares University, Noor 4641776489, Iran; ghasemrashidiyan@gmail.com (G.R.); azinfahim@gmail.com (A.F.)
2 Aquatic Animal Diseases Laboratory, Department of Aquaculture, National Institute of Oceanography and Fisheries, Suez P.O. Box 43511, Egypt; kh.shahin87@gmail.com
3 Department of Clinical Pathology, Faculty of Veterinary Medicine, Mansoura University, Mansoura P.O. Box 35516, Egypt; gehadelshopakey@mans.edu.eg
4 Department of Fish Diseases and Management, Faculty of Veterinary Medicine, Zagazig University, Zagazig P.O. Box 44511, Egypt; hhhosny@zu.edu.eg
5 Department of Aquatic Animals' Diseases and Management, Faculty of Veterinary Medicine, Benha University, Moshtohor P.O. Box 13736, Egypt; hiam.elabd@ftvm.bu.edu.eg
6 Department of Physiology, Institute for Biological Research "Siniša Stanković", National Institute of Republic of Serbia, University of Belgrade, 11060 Belgrade, Serbia; prokssima@gmail.com
7 Department of Chemical, Biological, Pharmaceutical and Environmental Sciences, University of Messina, 98166 Messina, Italy
* Correspondence: cfaggio@unime.it
† These authors contributed equally to this work.

**Abstract:** Medicinal plants are increasingly used in aquaculture owing to their beneficial impacts on the health status of farmed fish. The current study was conducted to investigate the effect of nutmeg (*Myristica fragrans*) extract on growth, immunity, antioxidant parameters, and resistance of common carp (*Cyprinus carpio*) against *Aeromonas hydrophila*. In addition, in vitro antibacterial activity of the skin mucus of fish fed on nutmeg extract was evaluated against three major fish pathogenic bacteria through the standard disk diffusion method. Fish (17.27 ± 0.11 g) were divided into four groups and fed on experimental diets containing different levels of nutmeg extract, including zero (control), 0.5% (M1), 1% (M2), and 2% (M3) per kg diet. Results showed that nutmeg significantly enhanced growth parameters after a four-week feeding trial. Feed conversion ratio was remarkably reduced with the lowest value reported for the M3 group, whereas weight gain was notably increased in M2 and M3. No significant effect was found on the hematological profile, including mean corpuscular hemoglobin concentration, mean corpuscular hemoglobin, mean corpuscular volume, and hematocrit, while the highest levels of red blood cells and white blood cells were found in the M3 group. Stress biomarkers, including glucose and cortisol, were the lowest in the M3 group. Serum and skin mucus immunological and antioxidant parameters were significantly higher in M3, followed by M2, where the highest resistance was also observed. In addition, skin mucus samples effectively inhibited *Streptococcus iniae*, *Yersinia ruckeri*, and *Aeromonas hydrophila*. Overall, the present results suggest that dietary nutmeg (20 g/kg diet) could be used as a growth promotor and immunostimulant in common carp.

**Keywords:** *Myristica fragrans*; medicinal herbs; growth performance; antibacterial activity; immunity; common carp

## 1. Introduction

Aquaculture supplies more than 60% of the world's food fish, and its further expansion seems inevitable. In intensive aquaculture, fish are fed only commercial feeds, and thus functional diets can contribute to the sustainability of the industry by increasing growth and heightening immunity.

Feed additives have been evaluated for the improvement of feed utilization, growth performance, immune responses, and resistance against various pathogens [1–4]. Natural feed additives, such as medicinal plant extracts, have been demonstrated to be efficient in reducing the need for chemical treatments or antibiotics, decreasing metabolic waste production, and improving fillet quality [5–13]. Thus, medicinal plants have been widely studied to discover their potential use in aquaculture.

Nutmeg (*Myristica fragrans*) is a dark-leaved evergreen tree cultivated for two spices derived from its seed (nutmeg) and seed covering (mace). It is also a commercial source of essential oils and nutmeg butter [14]. Nutmegs have been extensively used in traditional medicine as antithrombotic, antitumor, and anti-inflammatory [15]. In addition, *M. fragrans* is a natural curative plant rich in many bioactive substances exhibiting antibacterial and antioxidant activities [16]. Various active compounds have been identified in the raw nutmeg from which mace lignan showed significant antibacterial effects against various bacterial pathogens, including *Streptococcus mutans, S. sanguis, S. salivarius,* and *Lactococcus casei* [17]. In a recent report, the antimicrobial activity of nutmeg essential oil was investigated through an in vitro disc diffusion method against Gram-positive bacteria (*Staphylococcus aureus, Listeria monocytogenes, Bacillus cereus*) and Gram-negative bacteria (*Klebsiella pneumoniae, Escherichia coli, Proteus vulgaris, Pseudomonas aeruginosa*) [14]. Recent analysis showed that the major constituents of *M. fragrans* are terpene hydrocarbons (pinene, sabinene, camphene, p-cymene, terpinene, phellandrene, myrcene, and limonene) that constitute 60% to 80% of the oil, aromatic ethers (myristicin, elemicin, safrole, elemicin, and eugenol derivatives) representing 15% to 20% of the oil, and oxygenated terpenes (geraniol, linalool, and terpineol), which make up approximately 5% to 15% of the *M. fragrans* structure [18–20].

In aquaculture, nutmeg has been used for various purposes. Dietary inclusion of nutmeg noticeably improved weight gain, feed intake, and reduced mortality rates in *Clariusgarie pinus* [21] and *Cyprinus carpio* [22]. In addition, the anesthetic effect of *M. fragrans* was previously reported in *C. carpio* fingerlings following administration of 500 mg/L of nutmeg powder [15].

The present experiment was conducted on common carp (*Cyprinus carpio*) as an important freshwater species cultured in many countries that is listed among the top ten for both quantity and value [23]. The main aim of this experiment was to investigate the dietary effects of different levels of *M. fragrans,* including 0.5%, 1%, and 2% /kg diet, on growth, hematology, serum, mucosal immune response, hepatic function, antioxidant capacity, and survival of *C. carpio* challenged with *A. hydrophila*. There are several pathogenic bacteria that challenge the aquaculture industry, including *A. hydrophila.* This bacteria represent a worldwide issue infecting several freshwater and marine species, including Common carp [24], causing devastating economic losses. In addition, the in vitro bactericidal effect of skin mucus from *C. carpio* fed with *M. fragrans* against *A. hydrophila, Streptococcus iniae,* and *Yersinia ruckeri* was evaluated.

## 2. Materials and Methods

### 2.1. Preparation of Myristica Fragrans Extract

Dried nutmegs were provided from local herbal stores (Tehran, Iran) and minced using a commercial grinder (Parskhazar, Iran) until obtaining a fine powder. The powdered material was dissolved in absolute ethanol (ThermoFisher Scientific, Bedford, MA, USA) (2:1 *w/v* ratio) and sonicated for 1 h at 30 kHz/h in Bransonic ultrasonic baths (Thomas Scientific, Swedesboro, NJ, USA). The solvent was vaporized on a hot plate (VWR, Missis-

sauga, ON, Canada) at 40 °C under constant stirring overnight, and the resulting product was freeze dried and kept at 4 °C until use.

## 2.2. Preparation of Experimental Diets

A commercial fish feed (Faradaneh. Co, Shahr-e Kord, Iran) was used as the basal diet and supplemented with different levels of *M. fragrans* extract to form 4 feed groups including 0 (as control), 0.5% (M1), 1% (M2), and 2% (M3) per kg diet. In brief, the basal diet was mixed with distilled water until smooth dough was achieved, *M. fragrans* extract was added, and thoroughly mixed to form a homogenous paste. Before pelletizing the diet, 1 g of preheated gelatin was added to the paste, and then the prepared pellets were sieved and stored in plastic bags at 4 °C until use. The proximate composition of the experimental diets is shown in Table 1.

**Table 1.** Proximate composition of diets incorporated with zero (control), 0.5% (M1), 1% (M2), and 2% (M3) *Myristica fragrans* extract per kg diet.

| Proximate Composition of Diets (%) | Control | M1 | M2 | M3 |
|---|---|---|---|---|
| Crude protein | 32.41 | 32.8 | 33.31 | 32.82 |
| Crude lipid | 7.31 | 7.18 | 7.05 | 7.45 |
| NFE * | 30.15 | 31.35 | 31.41 | 30.88 |
| Ash | 12.55 | 12.86 | 12.1 | 12.35 |
| Moisture | 9.4 | 9.55 | 9.61 | 9.5 |
| Gross energy (MJ/kg) ** | 1196 | 1210 | 1217 | 1215 |

* Nitrogen free extract, ** gross energy calculated according to [25].

## 2.3. Fish Rearing Conditions

A total of 276 clinically healthy common carp (17.27 ± 0.11 g) were obtained from a local farm (Gilan, Iran) and acclimated for 3 weeks before starting the experiment. Following acclimation, fish were randomly distributed into 12 glass 200-L aquaria containing dechlorinated static water provided with air stones. The experiment included four treatments with triplicate tanks holding 23 fish/tank. A total of 20% of water was exchanged with fresh water, and uneaten food was removed by siphoning daily. Physicochemical parameters of rearing water, including temperature (23 ± 2 °C), pH (7.4 ± 0.18), dissolved oxygen (6.2 ± 0.71 mg/mL), and hardness ($CaCO_3$) (371 ± 34), were monitored during the experiment. The photoperiod was set as 12 h light:12 h dark throughout the experiment using artificial light. Fish were fed with the experimental diets at 2% of their respective body weight twice a day for a period of 4 weeks.

## 2.4. Sampling

Fish were off-feed for 24 h before sampling. Nine fish per treatment (three fish per tank) were randomly selected, euthanized by an overdose (1 g/L) of buffered MS-222 (Tricaine Methanesulfonate, Syndel, WA, USA), and used for blood and tissue sampling. Blood was collected using heparinized syringes from the caudal vein to evaluate hematological indices. For serum collection, another set of blood samples (three/tank) were collected using a non-heparinized syringe, immediately transferred into tubes without anticoagulant (International Scientific Supplies Ltd., Bradford, UK), incubated at room temperature (~25 °C) for 1 h, centrifuged at 3000 rpm for 10 min, and then clear serum samples were transferred to new 1.5 mL microcentrifuge tubes (Eppendorf, San Diego, CA, USA) and kept at −20 °C until further analysis.

Skin mucus samples were collected by gently rubbing individual fish placed inside plastic bags containing 5 mL of 50 mM NaCl (Sigma, Steinheim, Germany) from head to tail for approximately 1–3 min. 15 fish from each treatment (5 per tank) were euthanized as

mentioned above and used for mucus collection. The obtained mucus from each triplicate were pooled, immediately transferred to 50 mL sterile tubes, centrifuged using a 5810 R centrifuge (Engelsdorf, Germany) at 3500 rpm for 10 min at 4 °C, aliquoted into a 1.5 mL microcentrifuge tube, and stored at $-80$°C until use. Liver samples were collected to evaluate the antioxidant enzyme activities.

### 2.5. Growth Performance

Morphometric parameters were recorded at the end of the feeding trial (4 weeks) to evaluate growth performance, including weight gain (WG), specific growth rate (SGR), feed conversion ratio (FCR), and survival rate (SR), using the following equations:

$$WG \text{ (g)} = \text{Final weight} - \text{Initial weight};$$

$$FCR = \text{Total Feed Given (g)/Weight gain (g)};$$

$$SGR \text{ (\%/d)} = ([\text{Ln final wt (g)} - \text{Ln initial wt (g)}]/\text{days}) \times 100;$$

$$SR \text{ (\%)} = (\text{final numbers/initial numbers}) \times 100.$$

### 2.6. Analysis of Serum and Skin Mucus Immunological Parameters

Lysozyme activity was measured in serum according to the method of [26] using *Micrococcus luteus* (Sigma, M 3770, St. Louis, MI, USA). Alternative complement activity was determined based on the hemolysis of sheep red blood cells (SRBC) according to [27]. The volume of the sample yielding 50% hemolysis was determined and used to calculate the complement activity of the samples (value of ACH50 in units per mL) using the following equation:

$$ACH50 \text{ (U /mL)} = k \times 0.5 \times \text{(dilution factor)}$$

where "k" is the amount of serum in milliliters that caused 50% hemolysis.

The microprotein assay method was used to quantify plasma and mucus total immunoglobulin (total Ig) levels (C-690; Sigma, Aldrich, Saint Louis, MO, USA). 12% polyethylene glycol solution applied for Ig precipitation and total Ig level presented after subtracting protein content before and after precipitation [28]. Total protein in mucus samples was measured using commercial kits following the manufacturer's instructions (Total protein, Pars Azmun Co., Tehran, Iran).

### 2.7. Hematological Indices

To measure hemoglobin in blood samples (g/dL), the cyan-methhemoglobin method was followed [29]. A semi-automated spectrophotometer (BioTek 800TS, Agilent, Santa Clara, CA, USA) at $OD_{540}$ nm was used, and readings were compared to a standard curve to determine the amount of hemoglobin. In order to count the number of red blood cells (RBCs), blood was diluted 200 times with physiological saline in a red mélange pipette and counted using a hemocytometer under a CX23 Upright microscope (Olympus Canada Inc., Richmond Hill, ON, Canada). To count white blood cells (WBCs), blood was diluted 20 times in a white mélange pipette and counted using a hemocytometer under a microscope. Erythrocyte indices, including mean erythrocyte volume (MCV), mean erythrocyte hemoglobin (MCH), and mean erythrocyte hemoglobin concentration (MCHC), were determined using the following mathematical equations [30]:

$$MCHC = Hb \times 10/Hct \tag{1}$$

$$MCV = Hct \times 10/RBC \text{ (million)} \tag{2}$$

$$MCH = Hb \times 10/RBC \text{ (million)} \tag{3}$$

### 2.8. Assessment of Liver Enzymatic Activities and Oxidative Stress Parameters

Zellbio commercial kits (Zellbio, Veltinerweg, Germany) were used to assess superoxide dismutase (SOD) activity, catalase (CAT) activity, and malondialdehyde (MDA) concentration according to a previous study [31]. Alkaline phosphatase (ALP), and alkaline transaminase (ALT) were measured using commercial kits (abcam, Cambridge, England) according to the manufactures' instructions.

### 2.9. Anti-Bacterial Activities

### 2.9.1. Bacterial Strains

Strains of *Streptococcus iniae* (PTCC1887) and *Yersinia ruckeri* (PTCC1888) were obtained from the Persian Type Culture Collection (Tehran, Iran). *Aeromonas hydrophila* (RTCC1032) was provided by the Department of Aquatic Animal Health, Faculty of Veterinary Medicine, University of Tehran, Tehran, Iran. Bacteria were grown on tryptic soy agar (TSA) at 30 °C (for *S. iniae*) and 25 °C for *A. hydrophila* and *Y. ruckeri* for 24–48 h. A pure single colony of each bacterium was inoculated into Brain Heart Infusion broth (BHIB), followed by incubation at similar temperature and time mentioned before with shaking at 150RPM in a Biosan Shaker incubator ES-20 (Biosan, Riga, Latvia).

### 2.9.2. Disc Diffusion Test

Following the method of [32], 100 μL of the different bacterial suspensions, adjusted to a final concentration of 0.5 McFarland (~$1.5 \times 10^8$ CFU/mL) in phosphate buffered saline (PBS), were spread on Müller Hinton Agar plates (MHA, Sigma Aldrich, Saint Louis, MO, USA). Fifty microliters of mucus samples, collected from different fish treatments, and diluted in PBS (1:2 *v/v*), was poured on blank paper discs and air dried for 1 min. Discs were placed into the center of the MHA plates using sterile forceps and incubated at 28 °C for 24–48 h. The inhibition zone was measured using ImageJ software in millimeters.

### 2.10. Experimental Infection

The remaining fish (*n* = 45) from each treatment at the end of the feeding trial were randomly redistributed into new triplicate tanks to perform experimental infection. Fish were initially anesthetized using buffered MS-222 (85 mg/L), then intraperitoneally injected with 100 μL of $1.3 \times 10^8$ CFU/mL of *A. hydrophila* in PBS (stain RTCC1032). Fish were kept for 2 weeks, examined 3 times per day, fed ad libitum on the same diet twice daily, moribund, and recent mortalities were removed, and posterior kidney swabs were used for bacterial recovery on TSA plates incubated at 25 °C.

### 2.11. Statistical Analysis

Statistical analysis of data after checking the normality of their distribution by the Kolmogorov–Smirnov test was performed using one-way analysis of variance (one-way ANOVA), and Duncan post-hoc test at 95% confidence level was performed. Kaplan–Meier Survival Analysis (GraphPad V9.0, San Diego, CA, USA) with Log Rank pairwise comparisons (Mantel–Cox) were used for analysis of cumulative mortality data. The level of significance was $p < 0.05$.

## 3. Results

### 3.1. Growth Performance

The results revealed that fish fed with supplemented diets expressed higher overall growth performance compared to the control group, as shown in Table 2. Individuals from the M3 group showed the significantly highest values for growth parameters ($p < 0.05$) including final weight, weight gain, and specific growth rate compared to other treatments. The lowest feed conversion ratio was recorded in the M3 group, while other treatments showed no significant difference when compared to the control. No significant difference ($p > 0.05$) was found between the M1 and control groups for all parameters measured. No mortality was recorded during the experiment among the individuals from all treatments.

**Table 2.** Growth performance of *C. carpio* fed four diets incorporated with zero (control), 0.5% (M1), 1% (M2), and 2% (M3) of *Myristica fragrans* for four weeks.

| Parameters | Control | M1 | M2 | M3 |
|---|---|---|---|---|
| Initial weight (g) | 17.34 ± 0.11 a | 17.32 ± 0.05 a | 17.22 ± 0.05 a | 17.20 ± 0.01 a |
| Final weight (g) | 30.94 ± 1.42 b | 36.06 ± 1.51 ab | 40.96 ± 2.26 a | 42.29 ± 2.33 a |
| Weight gain (g) | 13.60 ± 1.32 b | 18.75 ± 1.47 ab | 23.74 ± 2.29 a | 25.09 ± 2.32 a |
| FCR | 2.03 ± 0.11 b | 1.59 ± 0.13 ab | 1.54 ± 0.17 ab | 1.19 ± 0.09 a |
| SGR (%/d$^{-1}$) | 1.03 ± 0.07 b | 1.31 ± 0.07 ab | 1.54 ± 0.10 a | 1.60 ± 0.10 a |
| SR (%) | 100 ± 0.00 a | 100 ± 0.00 a | 100 ± 0.00 a | 100 ± 0.00 a |

WG: weight gain; SGR: specific growth rate; FCR: feed conversion ratio; SR: survival rate. Values are presented as mean ± SE. Values with different letters within the same row are significantly different ($p < 0.05$).

### 3.2. Hematological Indices

Results showed no significant effect of *M. fragrans* supplementation on MCHC, MCV, and hematocrit among all groups. A remarkable increase in RBCs and WBCs was noticed in M3, while M2 and M1 showed no statistical difference ($p > 0.05$) compared to the controls. Hemoglobin was the highest in M3, followed by M2, where both were significantly different from that of the control group ($p < 0.05$). Statistical analysis showed no significant difference ($p > 0.05$) between M2 and M1, and M1 and the control group, respectively (Table 3).

**Table 3.** Hematological indices of *C. carpio* fed basal diet incorporated with zero (control), 0.5% (M1), 1% (M2), and 2% (M3) of *Myristica fragrans* for four weeks.

| Parameters | Control | M1 | M2 | M3 |
|---|---|---|---|---|
| MCHC | 43.04 ± 4.0 a | 41.84 ± 1.85 a | 45.98 ± 1.81 a | 53.35 ± 2.54 a |
| MCH | 42.84 ± 0.8 a | 33.36 ± 1.13 b | 45.73 ± 1.96 a | 42.44 ± 2.94 a |
| MCV | 101.07 ± 8.7 a | 79.86 ± 2.29 a | 99.41 ± 0.59 a | 79.53 ± 3.97 a |
| RBCs (×10$^6$/mm$^3$) | 2.44 ± 0.1 b | 3.09 ± 0.10 ab | 2.68 ± 0.08 bc | 3.33 ± 0.20 a |
| WBCs (×10$^3$/mm$^3$) | 6.31 ± 0.3 b | 6.41 ± 0.17 b | 7.19 ± 0.19 ab | 7.50 ± 0.17 a |
| Hematocrit (%) | 24.47 ± 1.0 a | 24.63 ± 0.26 a | 26.60 ± 0.64 a | 26.37 ± 1.04 a |
| Hemoglobin (g dL$^{-1}$) | 10.46 ± 0.6 c | 10.30 ± 0.40 bc | 12.21 ± 0.29 b | 14.02 ± 0.16 a |

Values are presented as mean ± SE. Values with different letters within the same row are significantly different ($p < 0.05$).

### 3.3. Serum and Mucus Immunological Parameters

Serum immunological responses (total Ig, lysozyme activity, and alternative complement hemolytic activity) are represented in Figure 1. The most significant increase in total Ig level was noticed in M3, followed by M2 treatments ($p < 0.05$). M1 was not significantly different from the control group ($p > 0.05$). Similar results were recorded for lysozyme activity. Analysis of complement activity showed that the M3 treatment recorded the highest value, while other treatments were not statistically different from the control group ($p > 0.05$).

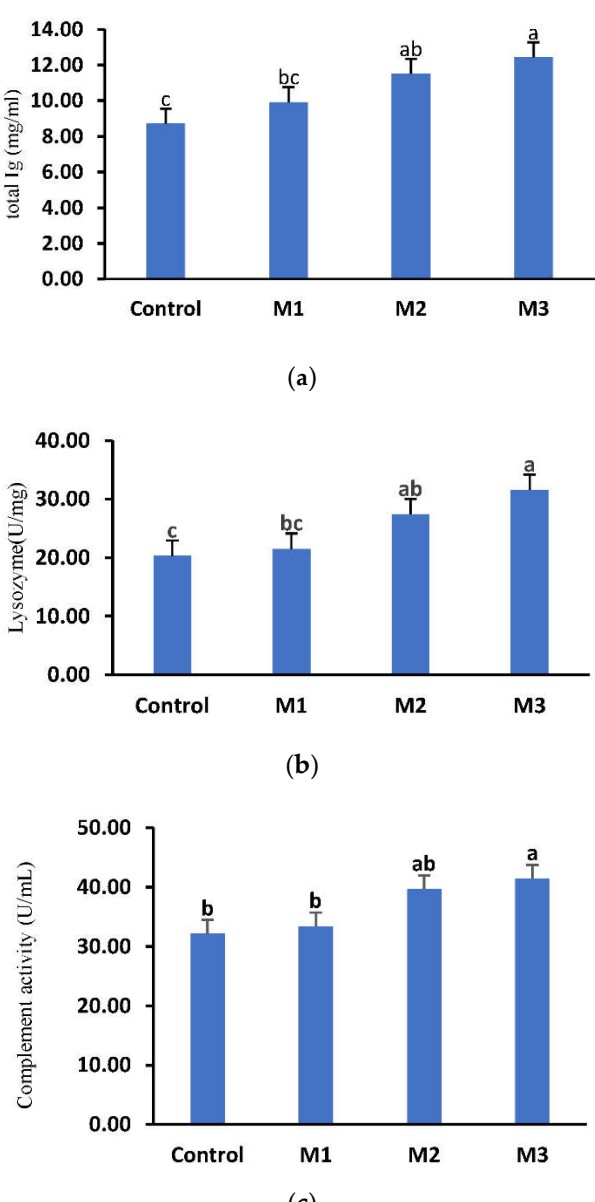

**Figure 1.** Serum immunological responses (total Ig (**a**), Lysozyme (**b**), and complement activity (**c**)) of *C. carpio* fed incorporated diet with zero (control), 0.5% (M1), 1% (M2), and 2% (M3) of *Myristica fragrans* for four weeks. Values are presented as the mean ± SE. Different letters indicate significant difference ($p < 0.05$).

### 3.4. Biochemical Parameters

Fish that received a *M. fragrans* supplemented diet showed a significant increase in total protein, albumin, and globulin levels when compared to those of the control group ($p < 0.05$), except for globulin records of M2 and M1 groups that did not statistically differ from the control group. Triglyceride, cholesterol, glucose, and cortisol were significantly lower in all treatments compared to the control group (Table 4).

**Table 4.** Serum biochemical parameters of *C. carpio* fed diet incorporated with different levels of zero (control), 0.5% (M1), 1% (M2), and 2% (M3) of *Myristica fragrans* for four weeks.

| Treatments | Control | M1 | M2 | M3 |
|---|---|---|---|---|
| Total protein (g dL$^{-1}$) | 1.80 ± 0.01 b | 3.03 ± 0.08 a | 3.10 ± 0.27 a | 3.18 ± 0.16 a |
| Albumin (g dL$^{-1}$) | 1.06 ± 0.12 b | 1.40 ± 0.22 ab | 1.88 ± 0.20 a | 1.91 ± 0.14 a |
| Globulin (g L$^{-1}$) | 0.75 ± 0.12 b | 1.63 ± 0.29 ab | 1.23 ± 0.20 ab | 1.27 ± 0.05 a |
| Triglyceride (mg dL$^{-1}$) | 183.34 ± 2.67 a | 167.26 ± 1.61 b | 160.60 ± 1.70 b | 166.35 ± 1.87 b |
| Cholesterol (mg dL$^{-1}$) | 164.33 ± 3.48 a | 154.70 ± 1.89 ab | 144.14 ± 2.01 bc | 138.63 ± 3.13 c |
| Glucose (mg dL$^{-1}$) | 68.91 ± 2.05 a | 64.20 ± 1.68 ab | 58.60 ± 2.64 bc | 52.60 ± 0.85 c |
| Cortisol (nmol L$^{-1}$) | 53.77 ± 0.98 a | 42.81 ± 1.38 b | 36.02 ± 1.94 c | 32.55 ± 1.05 c |

Values are presented as the mean ± SE. Values with different letters within the same row are significantly different ($p < 0.05$).

Analysis of liver enzyme activities showed that only ALT values were significantly lower in the M3 treatment compared to the control group (Table 5).

**Table 5.** Activity of liver enzymes of *C. carpio* fed a basal diet incorporated with different levels of zero (control), 0.5% (M1), 1% (M2), and 2% (M3) of *Myristica fragrans* for four weeks.

| Treatments | Control | M1 | M2 | M3 |
|---|---|---|---|---|
| ALT (U mL$^{-1}$) | 12.01 ± 0.71 a | 11.18 ± 0.80 ab | 9.41 ± 0.38 ab | 9.06 ± 0.34 b |
| AST (U mL$^{-1}$) | 21.04 ± 1.66b a | 19.66 ± 1.11 a | 17.88 ± 0.62 a | 16.61 ± 0.31 a |
| ALP (U mL$^{-1}$) | 25.68 ± 1.54 a | 23.59 ± 0.31 ab | 20.43 ± 0.59 a | 20.12 ± 0.81 a |

Values are presented as mean ± SE. Values with different letters within the same row are significantly different ($p < 0.05$).

No remarkable changes were detected in SOD values between the different groups, while CAT activity scored the significantly highest value in M3 treatment compared to other treatments. Malondialdehyde (MDA) showed no significant difference among treatments; however, individuals from M3 showed lower levels in comparison to the individuals from other groups (Figure 2).

*3.5. Skin Mucus Antibacterial Activity*

The antibacterial effects of fish mucus samples from different treatments are shown in Figure 3. Higher bactericidal activity was recorded for the *C. carpio* mucus that received experimental diets supplemented with nutmeg compared to that of control fish. Significantly greater inhibition zones were found for mucus samples from M3 treatment against the 3 tested pathogens compared to the other treatments. *A. hydrophila* was the lowest sensitive, while *S. iniae* was the most sensitive across all treatments.

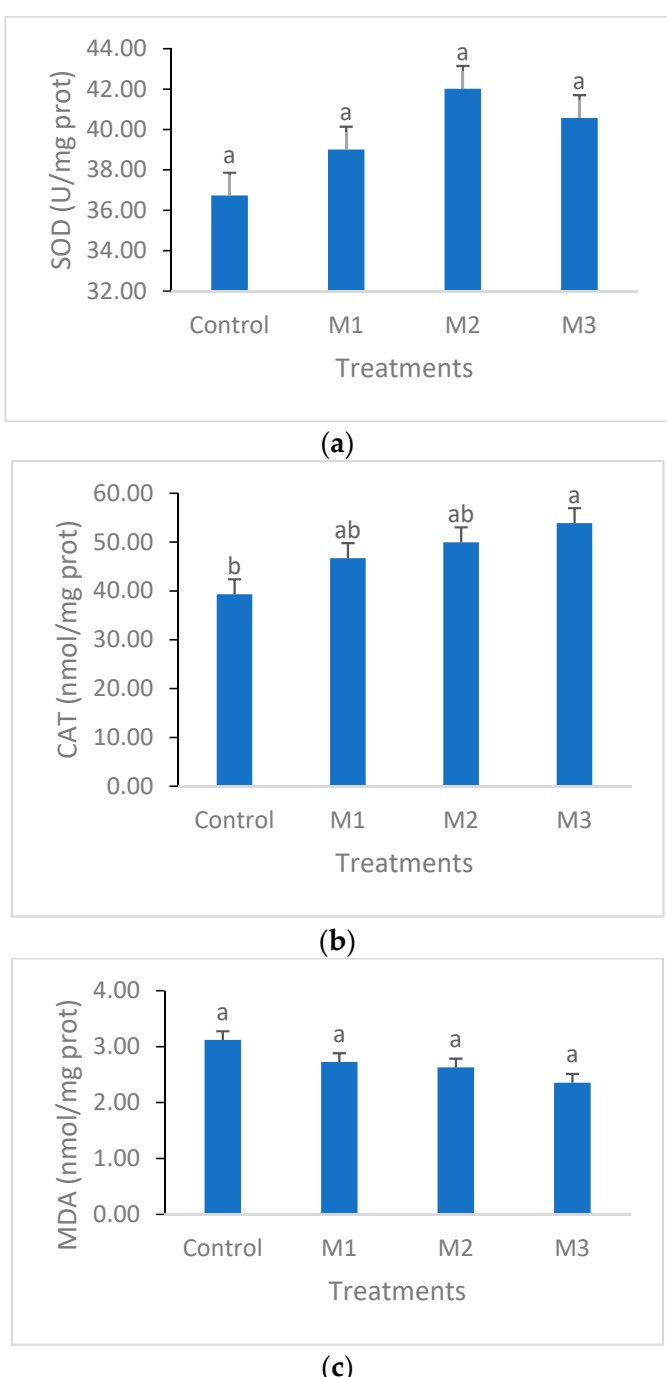

**Figure 2.** Oxidative stress parameters of SOD (**a**), CAT (**b**), and MDA (**c**) of *C. carpio* fed a basal diet incorporated with zero (control), 0.5% (M1), 1% (M2), and 2% (M3) of *Myristica fragrans* for four weeks. Values are presented as the mean ± SE. Different letters indicate significant differences ($p < 0.05$).

*3.6. Bacterial Challenge*

Following two weeks of experimental challenge with *A. hydrophila*, fish that received a diet incorporated with 2% *M. fragrans* extract (M3) showed significantly higher survival (68.8%) compared to that of the control (46.6%). No significant difference ($p > 0.05$) was found between survival of fish in M2 (57.7%) and M1 (55.5%), while both showed a significantly higher survival rate compared to the control group ($p < 0.05$) (Figure 4).

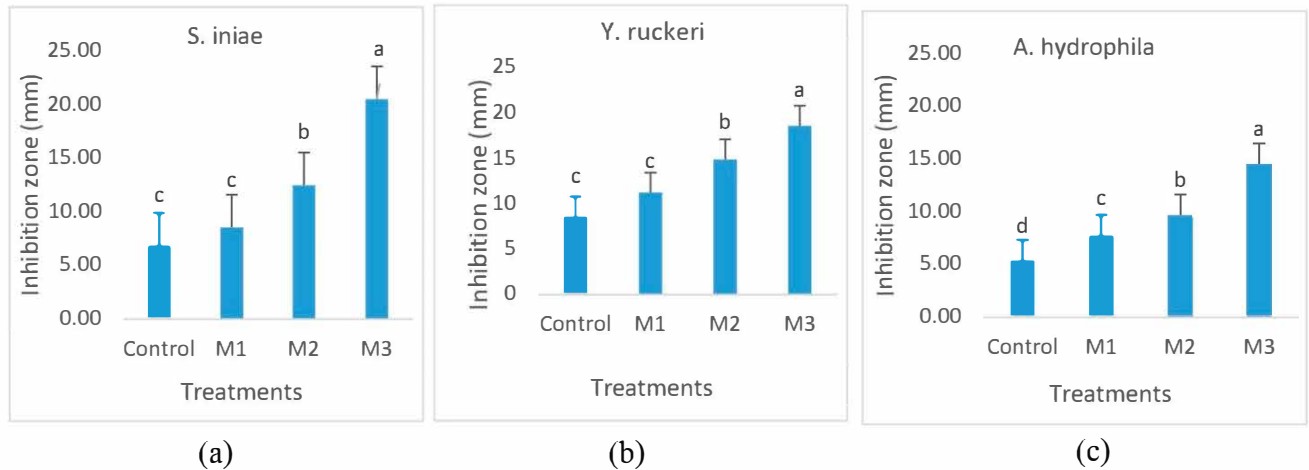

(a)  (b)  (c)

**Figure 3.** Antibacterial effects of skin mucus of *C. carpio* fed a basal diet incorporated with zero (control), 0.5% (M1), 1% (M2), and 2% (M3) of *Myristica fragrans* for four weeks against *Streptococcus iniae* (**a**), *Yersinia ruckeri* (**b**), and *Aeromonas hydrophila* (**c**) using the standard disk diffusion assay. Values were measured via ImageJ software in millimeters. Different subscripts indicate significant differences ($p < 0.05$).

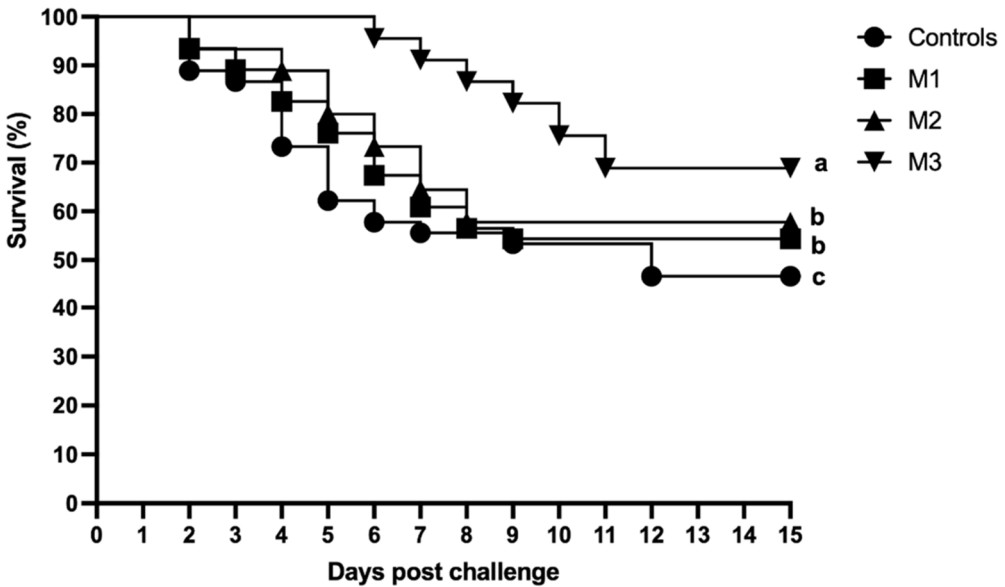

**Figure 4.** Survival rate of *C. carpio* fed a basal diet incorporated with zero (control), 0.5% (M1), 1% (M2), and 2% (M3) of *Myristica fragrans* following experimental challenge with *A. hydrophila* ($1.3 \times 10^8$ CFU mL$^{-1}$) for 15 days. Each curve represents the mortality mean of 3 tanks/treatment. Different letters indicate significant differences ($p < 0.05$).

## 4. Discussion

Aquaculture is one of the fastest growing sectors. Natural feed additives have proven to be efficacious and aid in reducing the need for medicinal treatments in aquaculture [33–35]. While there are great efforts to make the aquaculture industry sustainable, a wide range of microbial infections constantly leads to huge economic losses due to the emerging resistance to the commonly used antimicrobial drugs in aquaculture and the few available approved drugs to counteract infections in diseased fish. Thus, there is an urgent need to find alternatives to antimicrobials to control such significant pathogens in fish farms. Herein, we evaluated the effects of *M. fragrans* on growth, blood picture, immune status,

liver function, antioxidant capacity, survival of *C. carpio* challenged with *A. hydrophila*, and its in vitro antibacterial activity against some bacterial infections.

In our study, dietary supplementation with *M. fragrans* noticeably enhanced the growth of *C. carpio,* where the highest growth parameter scores were recorded in treatment that received 2% (M3) of *M. fragrans*. This finding is in agreement with the study of Sodamola et al. [21], where the authors reported that *Myristica* feed additive enhanced growth in catfish (*C. gariepinus*). The 2.5% dietary inclusion of Nutmeg in the juvenile catfish diet resulted in increasing weight gain and a reduction in the mortality rate [21]. Additionally, 1% nutmeg extract significantly improved weight gain and conversion ratio in common carp [22]. The stimulation of growth following dietary supplementation with *M. fragrans* could be attributed to its ability to enhance digestion and absorption, and its role as a protective agent for feed protein contents [36]. Blood analysis is a vital physiological indicator of the entire body condition and can be used to assess the health status and animal welfare [37]. Dietary supplementation with *Myristica* revealed a variable hematological response, with the remarked improvement in RBCs, WBCs, and Hb values for the group that received 2% of the extract. Similarly, Elabd et al. [38] reported that Moringa herbal incorporation enhanced hematological indices of Nile tilapia. The effect of *Myristica* in our study may be due to its ability to elicit an antioxidant response in RBC, which could protect their membranes from hemolysis and induce an elevation in the RBC count in the *Myristica*-enriched group.

Fish mucus is rich with many immune-related molecules, such as lysozymes, immunoglobulins, phosphatases, and proteins [39,40]. Lysozyme is a strong constituent of the innate immune system owing to its antibacterial efficacy. One of the mucus metabolites is protease, which is considered a vital biomarker for evaluation of both fish physiological and immunological responses. Protease commonly reacts to various challenges, such as bacterial infection [41,42]. In the current study, carp that received 2% of *Myristica* extract/kg diet showed the most significant improvement in immunological parameters. The immunostimulatory effect of *Myristica* can be attributed to augmenting the immune response and elevation of the immune biomarkers [42].

The liver is an important organ in fish and is responsible for storage, binding, and detoxification activities [43,44]. The activity of liver enzymes (ALT, ASP, and ALP) is related to protein metabolism and can provide insight into possible organ damage and general health status of the whole organism. Fish fed with *Myristica* had lower activities of liver enzyme, indicating that *Myristica* can display hepatoprotective efficacy, which modulates liver enzymes and protects liver tissue against physiological stress. This finding was in line with other studies that showed *Myristica* had a hepatoprotective influence and potent antioxidant activity owing to richness of phenolic contents and high free radical scavenging properties [45] and was able to suppress liver inflammation and lipid synthesis, which is represented by a noticeable improvement in liver function and blood lipid levels in mice [46].

The primary sign of freshwater fish oxidative distress is the alterations of the vital antioxidant enzymes evidenced by a decrease in CAT and SOD, and an increase in MDA [47]. Severe oxidative stress generates more reactive oxygen species (ROS), which have a major role in DNA, protein, and other cellular damage and increase lipid peroxidation [48,49]. A recent study highlighted the strong antioxidant effect of *Myristica* supplementation on SOD and CAT in rats [50], and its ability to prevent lipid peroxidation (MDA elevation) due to richness in antioxidant components [16]. Even though we presumed that *Myristica* supplementation would improve antioxidant status and lower lipid peroxidation process (MDA concentrations) in the liver of common carp, we have reported only a significant increase in CAT activity in the group with the highest dose of *Myristica* extract.

Based on the in vitro disc diffusion findings, *M. fragrans* exhibited potent antibacterial activity against *Streptococcus iniae*, *Yersinia ruckeri*, and *A. hydrophila*. Our findings were in agreement with a previous study in which nutmeg seeds exhibited antimicrobial activity against *Bacillus subtilis*, *Staphylococcus aureus,* and *Shigella dysenteriae* [51,52]. Additionally,

our results were in line with the study of [14] who reported that the extract of ethyl acetate of nutmeg has a strong inhibitory activity against Gram-positive bacteria (*Streptococcus species*) and three Gram-negative bacteria (*Aggregatibacter actinomycetemcomitans*, *Porphyromonas gingivalis*, and *Fusobacteriu mnucleatum*) with a minimum inhibitory concentration (MIC) ranging from 0.625 to 1.25 mg/mL. In another study, nutmeg was found to be rich with essential oil that decreases the growth and survival of *Yersinia enterocolitica* and *Listeria monocytogenes* in broth culture [53]. Additionally, a recent report highlighted the antimicrobial activity of nutmeg essential oil using a disc diffusion method on Gram-positive bacteria (*Staphylococcus aureus*, *Bacillus cereus*, *B. luteus*, *Listeria monocytogenes*) and Gram-negative bacteria (*Escherichia coli*, *Klebsiella pneumoniae*, *Pseudomonas aeruginosa*, *Proteus vulgaris*) [14,53].

## 5. Conclusions

The current study highlights the potential use of *Myristica fragrance* as a growth promotor, immune stimulant, antioxidant, and antibacterial agent in *C. carpio*. A higher inclusion level of the plant was found to be more effective, and therefore 2% incorporation into the diet is recommended. Hence, future research is warranted to investigate immune-related gene expression following the administration of feed supplemented with *M. fragrance*, as well as to test its effect on other important cultured fish species.

**Author Contributions:** Conceptualization G.R., K.S., H.H.M., H.E., G.E.E., A.F., M.D.P. and C.F.; methodology, G.R. and A.F.; software, G.R., M.D.P. and K.S.; validation, G.R., H.H.M., K.S., H.E., G.E.E., A.F., M.D.P. and C.F.; formal analysis, G.R., M.D.P. and K.S.; investigation, G.R., M.D.P., K.S., A.F., H.H.M., H.E. and C.F.; resources, G.R., H.H.M., K.S., H.E., G.E.E., A.F., M.D.P. and C.F.; data curation, G.R., M.D.P. and K.S.; writing—original draft preparation, G.R., K.S., H.H.M., H.E., G.E.E., A.F., M.D.P. and C.F.; writing—review and editing, G.R., M.D.P., K.S., H.H.M. and H.E.; visualization, G.R., M.D.P. and K.S.; supervision, G.R., K.S. and C.F.; project administration, G.R.; funding acquisition, G.R., K.S., H.H.M., H.E., G.E.E., A.F., M.D.P. and C.F. All authors have read and agreed to the published version of the manuscript.

**Funding:** This research received no external funding.

**Institutional Review Board Statement:** Not applicable.

**Informed Consent Statement:** Not applicable.

**Data Availability Statement:** Request to corresponding author of this article.

**Conflicts of Interest:** The authors declare no conflict of interest.

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
