# Peer review of "The Dietary Effects of Nutmeg (Myristica fragrans) Extract on Growth, Hematological Parameters, Immunity, Antioxidant Status, and Disease Resistance of Common Carp (Cyprinus carpio) against Aeromonas hydrophila"

_jmse, doi:10.3390/jmse10030325_

Round 1

Reviewer 1 Report

Dear authors, below are some specific comments on your paper:

  • in the Abstract, please don't use abbreviations (for example, FCR, MCHC, MCH..., write full terms);
  • in the Introduction, line 60, you are mentioning Matulyte et al., 2020 but in the reference list this is a reference number 49 - please, correct this;
  • Table 1 - is the value of 112.35 for Ash in M3 correct?;
  • in the Discussion, line 390, you are missing a reference (...in line with the study of) who reported...).

Author Response

Reviewer 1:

  1. in the Abstract, please don't use abbreviations (for example, FCR, MCHC, MCH..., write full terms);

Response:

All abbreviations were replaced by full words in the amended version of the MS.

  1. in the Introduction, line 60, you are mentioning Matuly te et al., 2020 but in the reference list this is a reference number 49 - please, correct this;

Response:

 The missing reference was added in the reference list of the amended MS,

  1. Table 1 - is the value of 112.35 for Ash in M3 correct?

Response:

We apologize for the typo. The correct value was added in the amended MS which is 12.35.

  1. in the Discussion, line 390, you are missing a reference (...in line with the study of) who reported...).

Response:

 The missing reference was added in the amended MS (now reads as reference “14”).

Reviewer 2 Report

In this article we can find a series of experiments that confirm the benefits of supplementing the Cyprinus carpio diet with nutmeg extract. In general, the article seems correct to me, although there are a number of doubts that I would like to see resolved before fully accepting it for publication.

Material and methods:

2.1: Has the nutmeg purchased at the local store been characterized in the laboratory to compare the chemical composition with the composition of other nutmegs used in other studies?

2.10: Has the mortality test only been performed once? How can you determine that the results will be the same when repeated a second time?

Results:

3.2: How could it be explained that only M1 shows a significantly lower MCH than the rest of the treatment concentrations? How relevant is this parameter to explain the benefits/harms of this compound?

3.4: ALT and ALP data is missing in Graph 2, although they are mentioned in materials and methods section 2.8.

3.5: I believe that the statistical differences that are represented in figure 3 are not very clear, since there are bars that seem similar, however they are represented as statistically different.

3.6: I think that in the introduction there should be a brief explanation of why A. hydrophila is chosen for the experimental infections if in the disk diffusion assay it is observed that S. iniae is the bacterium that shows the most sensitivity to this treatment.

Figure 4: The y-axis represents percent survival, not probability of survival.

Line 330: M3 label is lowercase.

Line 331: In the units you have to choose between CFU/ml or CFU ml-1, all together (CFU/ml-1) is incorrect.

Discussion:

Line 346: The 2 % of the treatment corresponded to M3, not to M2.

Line 390: There is a symbol ) that does not have to be there.

Missing data in the results are not discussed in this section either.

Author Response

Reviewer 2:

Material and methods:

2.1: Has the nutmeg purchased at the local store been characterized in the laboratory to compare the chemical composition with the composition of other nutmegs used in other studies?

Response:

No, we have not compared the nutmeg used in our study, which in fact is a commercial product, to other commercially available or other brands of nutmeg used in other studies. The main goal of our study to perform preliminary testing of nutmeg in carp and we will do further research in the future, and we will consider comparing the characters and effect of nutmeg from various sources as suggested. 

2.10: Has the mortality test only been performed once? How can you determine that the results will be the same when repeated a second time?

Response:

- In our experiment we run the infection trial once using triplicate tanks. We believe repeating infection trial twice is challenging (time consuming, expensive …etc).

- Regard the reproducibility, we previously performed dose response tests (pre-challenge testing) in our lab using different doses of the same A. hydrophila strain used in our main experiment in naiive carps to optimize the experiment (target 60-70% mortality) and we can confirm the current results is highly comparable to the previous data.

Results:

3.2: How could it be explained that only M1 shows a significantly lower MCH than the rest of the treatment concentrations? How relevant is this parameter to explain the benefits/harms of this compound?

Response:

We cannot rule out one single reason but may be the low dose of Mystrica can be associated with decrease size of RBCs inducing low MCH. However, we need further investigation to uncover that finding.

From both our finding and the few published research on using Mystrica in aquaculture, there is no harm or side effects from using that plant.  

3.4: ALT and ALP data is missing in Graph 2, although they are mentioned in materials and methods section 2.8.

Authors response:

Data for ALT, AST and ALP are now listed in Table 5 in the amended MS (attached below)

Table 5. Activity of liver enzymes of C. carpio fed a basal diet incorporated with different levels of zero (control), 0.5% (M1), 1% (M2), and 2% (M3) of Myristica fragransfor four weeks.

Treatments

Control

M1

M2

M3

ALT (U.ml-1)

12.01±0.71 a

11.18±0.80 ab

9.41±0.38 ab

9.06±0.34 b

AST (U.ml-1)

21.04±1.66b a

19.66±1.11 a

17.88±0.62 a

16.61±0.31 a

ALP (U.ml-1)

25.68±1.54 a

23.59±0.31 ab

20.43±0.59 a

20.12±0.81 a

3.5: I believe that the statistical differences that are represented in figure 3 are not very clear, since there are bars that seem similar, however they are represented as statistically different.

Response:

Statistical significance symbols were amended in the new version of the MS. New figure was added.

                         (a)                                               (b)                                                           (c)        

3.6: I think that in the introduction there should be a brief explanation of why A. hydrophila is chosen for the experimental infections if in the disk diffusion assay it is observed that S. iniae is the bacterium that shows the most sensitivity to this treatment.

Response:

We have added the following lines to the manuscript that reads:

There are several pathogenic bacteria that challenge aquaculture industry among which A. hydrophila is a worldwide issue infecting several freshwater and marine species including common carp [24] causing devastating economic losses.

Figure 4: The y-axis represents percent survival, not probability of survival.

Response:

 amended in the new version of the MS.

Line 330: M3 label is lowercase.

Response:

 corrected in the amended MS.

Line 331: In the units you have to choose between CFU/ml or CFU ml-1, all together (CFU/ml-1) is incorrect.

Response:

 units were standardized in the amended MS. 

Discussion:

Line 346: The 2 % of the treatment corresponded to M3, not to M2.

Response:

Line 390: There is a symbol ) that does not have to be there.

Response

corrected in the amended MS.

Missing data in the results are not discussed in this section either.

Response:

 missing data was added in the amended MS that reads:

Liver is an important organ in fish which is responsible for storage, binding, and detoxification activities [43,44]. The activity of liver enzymes (ALT, ASP and ALP) is related to protein metabolism and can provide insight in possible organ damage and general health status of whole organism. Fish feed with Myristica had lower activities of liver enzyme, indicating that Myristica can displayed a hepato-protective efficacy which modulate the liver enzymes and can protect liver tissue against stress”